# Electromagnetic Torque Components Analysis of Ultra-High-Speed Permanent-Magnet Synchronous Motor for Fuel Cell Air Compressor

**Jiaming Zhou [1], Jinming Zhang [1,*], Fengyan Yi [2], Donghai Hu [3,*], Caizhi Zhang [4], Yanzhao Li [1], Zhiming Zhang [5], Guangping Wu [6] and Jinxiang Song [6]**

1   School of Intelligent Manufacturing, Weifang University of Science and Technology, Weifang 262700, China; jnlmhs@126.com (J.Z.); liyanzhao@wfust.edu.cn (Y.L.)
2   School of Automotive Engineering, Shandong Jiaotong University, Jinan 250357, China; yi_fengyan@163.com
3   School of Automotive and Traffic Engineering, Jiangsu University, Zhenjiang 212013, China
4   College of Mechanical and Vehicle Engineering, Chongqing University, Chongqing 400044, China; czzhang@cqu.edu.cn
5   School of Automotive Studies, Tongji University, Shanghai 201804, China; zhangzm@tongji.edu.cn
6   Zhongtong Bus Co., Ltd., Liaocheng 252000, China; pingguangwu1989@163.com (G.W.); zjksjx@163.com (J.S.)
*   Correspondence: zhangjinming032@126.com (J.Z.); 1000004735@ujs.edu.cn (D.H.)

**Abstract:** The ultra-high-speed electric air compressor (UHSEAC) is affected by the electromagnetic torque components of the ultra-high-speed permanent magnet synchronous motor (UHSPMSM) during wide-range speed regulation, resulting in intense speed fluctuation. Electromagnetic torque components are generated by the effects of permanent magnet field harmonics, stator slotting, and current harmonics. It is very important to conduct simulation comparisons and theoretical descriptions of different sources of pulsation factors. In this paper, firstly, the electromagnetic torque model of UHSPMSM with a rated speed of 80,000 rpm is constructed and verified by an experimental bench. Secondly, the electromagnetic torque components of UHSPMSM are extracted on the basis of the electromagnetic torque model. Finally, the electromagnetic torque components' characteristic law is investigated under different ultra-high-speed operating conditions. The results show that under ultra-high-speed operation, the frequency and amplitude of electromagnetic torque components become larger with increasing speed. And the amplitude of electromagnetic torque components becomes larger with increasing torque. This paper constructs the observation object of the high-frequency state observer and does the preliminaries for the design of the UHSEAC controller.

**Keywords:** fuel cell; ultra-high-speed electric air compressor; ultra-high-speed permanent magnet synchronous motor; electromagnetic torque components; amplitude–frequency characteristics

## 1. Introduction

With the development of high-power fuel cell systems in fuel cell vehicles [1–4], ultra-high-speed electric air compressors (UHSEAC) with high pressure ratio and flow rate have been widely studied [5,6]. However, UHSEAC need to be modulated in seconds, which excites the speed to produce intense fluctuations [7,8]. The speed fluctuation of the UHSEAC is generated by the influence of the ultra-high-speed permanent magnet synchronous motor (UHSPMSM) electromagnetic torque components [9,10]. Therefore, it is very important to study the electromagnetic torque components in order to maintain the stable torque output of UHSPMSM.

The permanent magnet synchronous motor (PMSM) consists of a wounded stator and rotor equipped with permanent magnets. The permanent magnet synchronous motor controller inputs the adjusted three-phase current to the wound stator to generate a rotating electromagnetic field. When the rotating magnetic field of the stator interacts with the permanent magnetic field of the rotor, electromagnetic torque is generated to drive the motor.

The working process of the permanent magnet synchronous motor involves the influence of the current, machinery, and magnetic field [11–14]. Scholars explored the mechanism affecting the torque output stability of PMSM from the above three parts, respectively. Zhou et al. [15] illustrated that torque pulsations due to air gap magnetic field variations limit the torque output stability performance of PMSM. Girgin et al. [16] described that the stator slots have an effect on the torque output stability of PMSM. Yamazaki et al. [17] analyzed the mechanism of torque ripple generated by the harmonic magnetic field in PMSM. Qu et al. [18] illustrated that harmonic currents can generate harmonic magnetic fields and can make the torque output of the PMSM poorly stabilized. Electromagnetic torque components are generated under the combined influence of permanent magnet field harmonics, stator slots, and harmonic currents. The above only analyzed the torque output stability of PMSM for a single factor among them, without considering the three influencing factors at the same time, which will lead to a too one-sided analysis.

The characteristics of electromagnetic torque components are one of the key points to study the torque output stability of UHSPMSM. The UHSPMSM is a complex coupling system [19–22]. Its torque output stability is closely related to electromagnetic parameters, mechanical parameters, and electronic control parameters. Peng et al. [23] investigated the effect of rotor permanent magnet width on torque pulsation in a PMSM. Xu et al. [24] weakened the low-order harmonics of cogging torque and pulsating torque by suppressing the harmonics of the rotor magnetic pole magnetic field. Caruso et al. [25] found that the cogging torque component could be eliminated by changing the shape acting on the rotor laminations. Knypiñski et al. [26] designed algorithms and software for permanent magnet synchronous motor rotor structure optimization to improve the output performance of permanent magnet synchronous motors. Jędryczka et al. [27] studied the effect of harmonic currents on torque pulsations of PMSM. Zhao et al. [28] investigated the effect of electronic control parameters on the torque output stability of PMSM. The above have only investigated the effect of medium-conventional parameters on torque output stability of PMSM in low-speed operation conditions. However, UHSPMSM have the characteristics of ultra-high-speed and low-torque. The poor control accuracy of UHSPMSM under ultra-high-speed operating conditions can easily lead to operating instability. Therefore, it is very important to investigate the characteristics of electromagnetic torque components in UHSPMSM under different ultra-high-speed operating conditions.

Based on this, the research framework of this paper is shown in Figure 1. Based on the current scholars only for the single-factor analysis of electromagnetic torque components, this paper comprehensively considers the permanent magnet field harmonics, stator slotting, and current harmonics to extract the electromagnetic torque components. Since they have only studied the effect of medium-conventional parameters on the torque output stability of PMSM, this paper analyzes the amplitude–frequency characteristics of UHSPMSM electromagnetic torque components under ultra-high-speed operating conditions. There is a simulation comparison and theoretical description of different sources of pulsation factors. The main innovations of this paper are as follows:

(1) The UHSPMSM electromagnetic torque components that combine the effects of permanent magnet field harmonics, stator slotting, and current harmonics are extracted.
(2) The rules of electromagnetic torque components' amplitude and frequency characteristics under different ultra-high-speed operating conditions are revealed.

The rest of this paper is organized as follows: Section 2 constructs the UHSPMSM electromagnetic torque model and verifies the accuracy of the model through experimental bench. Section 3 extracts the electromagnetic torque components under permanent magnet magnetic field harmonics, stator slotting, and current harmonics. Section 4 analyzes the electromagnetic torque components amplitude–frequency characteristics of UHSPMSM. Section 5 shows the main conclusions.

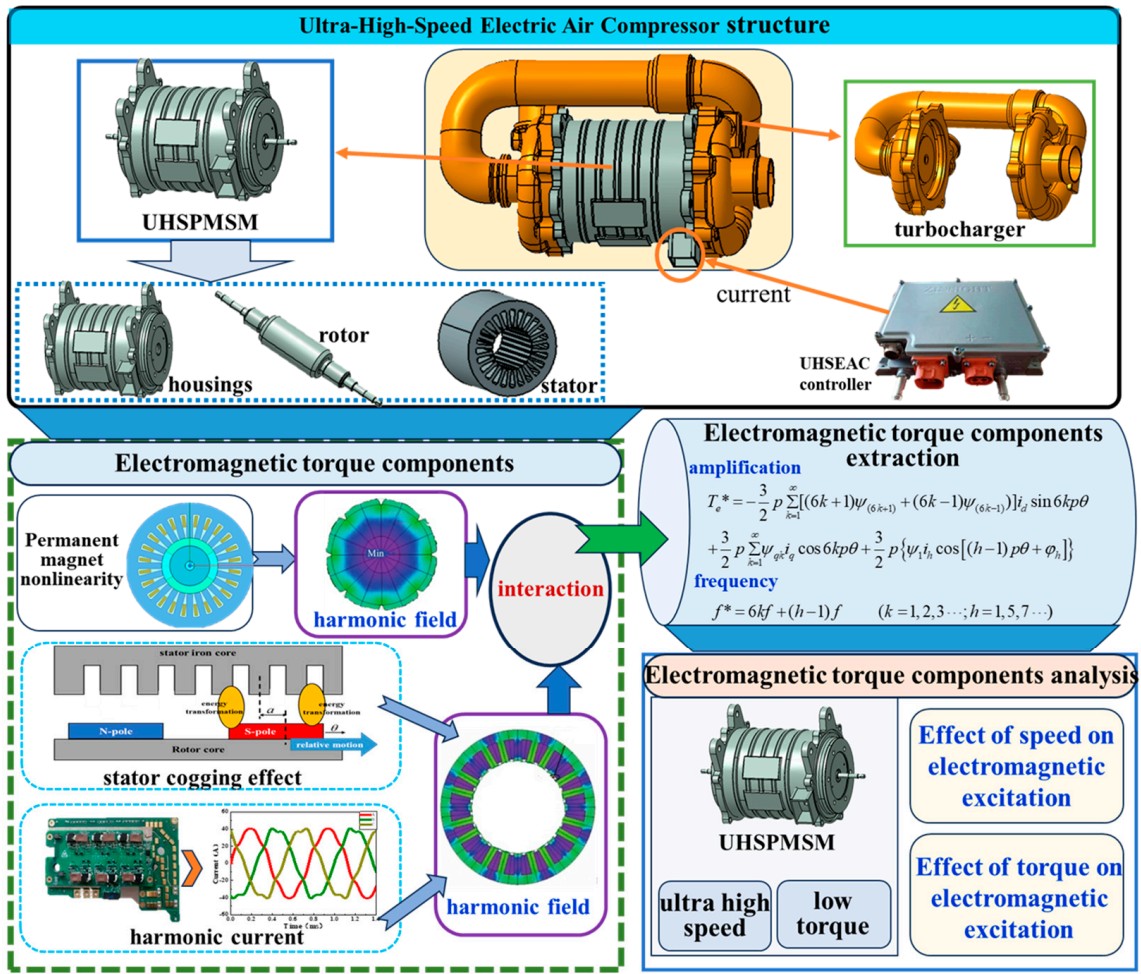

**Figure 1.** Research framework diagram of this paper.

## 2. Modeling and Experimental Verification of UHSPMSM

### 2.1. UHSEAC Structure

The whole structure of the UHSEAC is shown in Figure 1, which is mainly composed of the UHSPMSM and supercharger. Because the UHSPMSM has the advantages of high efficiency and small size [29,30], it used to be the power source. The UHSPMSM is mainly composed of the housing, stator, and rotor [31–33]. Slots are present on the inner surface of the stator, and three-phase excitation windings, a/b/c are erected in the slots. The three-phase excitation winding produces a rotating magnetic field with current input. The rotor mainly consists of a rotor shaft and permanent magnets. Each end of the rotating shaft is fastened to an impeller. The surface-mounted permanent magnet mounting method is used, which has the advantage of lower harmonic content. Then the model is constructed and simulated based on the parameters of UHSPMSM. The main parameters of UHSPMSM are shown in Table 1.

The UHSEAC controller adopts a typical vector control method and adopts the speed control mode [34,35]. The following of the target speed is guaranteed by calculating the difference between the target speed and the actual speed. After the command calculation is completed, the *d* and *q* axis stator current commands are determined based on the current speed. Then, the values of *d* and *q* axis stator voltages are determined under the action of the current PID regulator. A space vector pulse width modulation technique is used to generate a duty cycle signal that controls the switching of the thyristor, thus rotating the rotor.

**Table 1.** Main parameters of UHSPMSM.

| Parameter | Value | Unit |
|---|---|---|
| Rated power | 22 | kW |
| Rated voltage | 380 | V |
| Rated speed | 80,000 | rpm |
| Pole pairs | 1 | N/A |
| Number of slots | 24 | N/A |
| Stator inner diameter | 36.85 | mm |
| Stator outer diameter | 82 | mm |
| Stator length | 70 | mm |
| Stacking coefficient | 0.85 | N/A |
| Rotor inner diameter | 5.85 | mm |
| Rotor outer diameter | 35.85 | mm |
| Rotor shaft length | 70 | mm |
| Permanent magnet material | Arnold_Magnetics_N35EH_120C | N/A |
| Magnet thickness | 6 | mm |
| Magnet span | 60 | ° |
| Permanent magnet installation method | Surface mounted | N/A |
| Permanent magnet magnetizing direction | Radially magnetized | N/A |
| Permanent magnet material | NdFe35 | N/A |

*2.2. Electromagnetic Torque Modeling of UHSPMSM*

Ideally, the stator winding produces sinusoidally distributed currents under the control of the controller. The instantaneous values of the three-phase stator currents are set as $i_a$, $i_b$, $i_c$, respectively. Based on the winding method of the three-phase winding inside the UHSPMSM, the voltage equation in the three-phase stationary coordinate system is [36]:

$$\begin{pmatrix} U_a \\ U_b \\ U_c \end{pmatrix} = \begin{pmatrix} R & 0 & 0 \\ 0 & R & 0 \\ 0 & 0 & R \end{pmatrix} \begin{pmatrix} i_a \\ i_b \\ i_c \end{pmatrix} + p \begin{pmatrix} \psi_a \\ \psi_b \\ \psi_c \end{pmatrix} \tag{1}$$

where $U_a$, $U_b$, and $U_c$ are phase $a$, $b$, and $c$ voltages; $R$ is the stator phase winding resistance; $\psi_a$, $\psi_b$, and $\psi_c$ are the phase $a$, $b$, and $c$ magnetic fluxes; and $p$ is the differential operator.

The three-phase magnetic chain of the stator winding of UHSPMSM is:

$$\begin{pmatrix} \psi_a \\ \psi_b \\ \psi_c \end{pmatrix} = \begin{pmatrix} L_a & M_{ab} & M_{ac} \\ M_{ba} & L_b & M_{bc} \\ M_{ca} & M_{cb} & L_c \end{pmatrix} \begin{pmatrix} i_a \\ i_b \\ i_c \end{pmatrix} + \begin{pmatrix} \psi_{fa} \\ \psi_{fb} \\ \psi_{fc} \end{pmatrix} \tag{2}$$

where $L_a$, $L_b$, and $L_c$ are the stator three-phase winding self-inductance, respectively; $M$ represents the stator three-phase winding mutual inductance.

According to the theory of electromagnetism, the electromagnetic torque equation is:

$$T_e = \frac{1}{\omega}(\psi_a i_a + \psi_b i_b + \psi_c i_c) \tag{3}$$

where $\omega$ is the UHSPMSM speed and $T_e$ is the electromagnetic torque.

According to the Clark and Park transformation principle, the following transformation relations can be obtained:

$$\begin{pmatrix} i_d \\ i_q \\ i_0 \end{pmatrix} = \frac{2}{3} \begin{pmatrix} \cos\theta & \cos(\theta - \frac{2}{3}\pi) & \cos(\theta + \frac{2}{3}\pi) \\ -\sin\theta & -\sin(\theta - \frac{2}{3}\pi) & -\sin(\theta + \frac{2}{3}\pi) \\ \frac{1}{2} & \frac{1}{2} & \frac{1}{2} \end{pmatrix} \begin{pmatrix} i_a \\ i_b \\ i_c \end{pmatrix} \tag{4}$$

where $i_d$, $i_q$, and $i_0$ are the $d$, $q$, and 0 axis currents; $\theta$ is the angle between the $a$-phase axis and the $d$-axis; and the 3D matrix is the transformed matrix after performing the Clark and Park transforms.

The voltage equation for UHSPMSM in the $d$, $q$, and coordinate system is:

$$\begin{cases} U_d = Ri_d + \frac{d\psi_d}{dt} - \omega_r\psi_q \\ U_q = Ri_q + \frac{d\psi_q}{dt} + \omega_r\psi_d \end{cases} \tag{5}$$

The magnetic chain equation of UHSPMSM in the $d$ and $q$ coordinate system is:

$$\begin{cases} \psi_d = L_d i_d + \psi_{PM} \\ \psi_q = L_q i_q \end{cases} \tag{6}$$

where $\psi_{PM}$ is the permanent magnet magnetic chain, and $L_d$, $L_q$ are the $d$ and $q$ axis inductances.

Considering that the permanent magnets are mounted in a surface-mounted way, there are $L_d = L_q = L$. The electromagnetic torque equation in the synchronous rotating coordinate system is obtained as:

$$T_e = \frac{3}{2}p\psi_{PM}i_q \tag{7}$$

where $p$ is the pole pairs.

The main parameters in the electromagnetic torque model of UHSPMSM are shown in Table 2.

**Table 2.** Main parameters in the electromagnetic torque model of UHSPMSM.

| Parameter | Value | Unit |
|---|---|---|
| Permanent magnet chain, $\psi$ | 58 | mWb |
| Stator phase winding resistance, $R$ | 98 | m$\Omega$ |
| $d$ and $q$ axis inductances, $L_d/L_q$ | 0.25 | mH |
| Pole pairs, $p$ | 1 | N/A |

Based on the working mechanism and parameters of the UHSPMSM, the Maxwell 2D model of the UHSPMSM was built in ANSYS Electronics Desktop 2020R1 software.

### 2.3. Model Experimental Verification

(1)　Experimental device

In order to verify the accuracy of the UHSPMSM model, this paper built the UHSEAC external characteristics experimental bench, as shown in Figure 2. The experimental bench is mainly composed of an air supply system, testing equipment, power supply equipment, upper computer, wiring harness, and industrial chiller. The air supply system includes: UHSEAC and its controller, air filter, air outlet pipeline, electromagnetic back pressure valve. Testing equipment includes: temperature sensor, pressure sensor, flow differential pressure sensor. Power supply equipment includes: high-voltage DC power supply and low-voltage DC power supply.

(2)　Experimental method

In the experiment, different target speeds were set, and the data such as voltage and current in the controller of the UHSEAC were recorded. Firstly, fix the opening degree of electromagnetic back pressure valve as 36%. Secondly, open the data sending window of the software of the upper computer, set the target speed of 70,000 rpm. Thirdly, after the UHSEAC stops running, save the current and voltage from UHSEAC controller and high-voltage power supply. Finally, set the target speeds of 80,000 rpm and 90,000 rpm in turn, and repeat according to the above experimental steps.

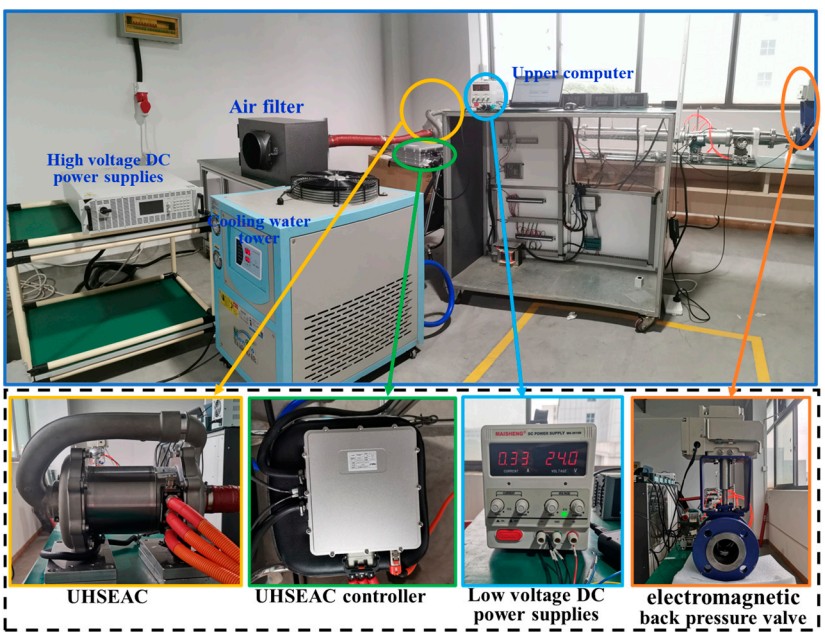

**Figure 2.** UHSEAC experimental bench diagram.

(3)    Experimental result

The experimental data are extracted from the experiment of external characteristics of UHSEAC when the speed is 70,000 rpm, 80,000 rpm, and 90,000 rpm under the condition that the opening degree of the back pressure valve is 36%. According to the current and voltage data of the controller of the UHSEAC, the electromagnetic torque of the actual UHSPMSM can be obtained through calculation. In the Maxwell 2D model of the UHSPMSM, the same three-phase current input as in the experiment was set, and finally, the electromagnetic torque data obtained from the experiment was compared with the simulation results. The maximum error between the simulation and the experimental results is 1.6%, as shown in Figure 3. The error range requirement is satisfied, thus verifying the accuracy of model.

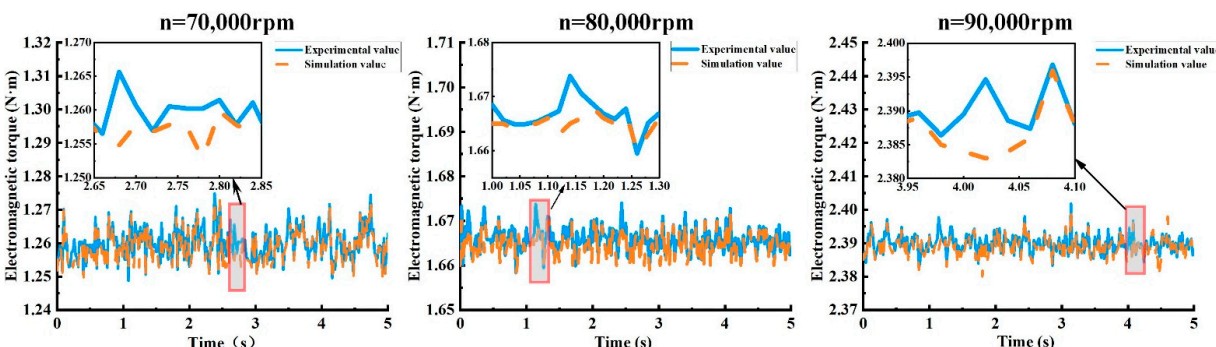

**Figure 3.** Comparison between simulated and experimental electromagnetic torque.

## 3. Extraction of Electromagnetic Torque Components for UHSPMSM

Electromagnetic torque components mainly consist of the following parts: ripple torque due to the nonlinear distribution of the magnetic field of the permanent magnets, cogging torque due to stator slotting, and pulsating torque due to current harmonics generated by the nonlinear characteristics of the UHSEAC controller.

(1)    Ripple torque

UHSPMSM have technical limitations due to the rotor permanent magnet manufacturing level and precision processes. The rotor permanent magnet magnetic field contains a

large magnetic field harmonic component. The harmonic magnetic field of the permanent magnet contains a series of spatial harmonics in the air gap that vary with position. This leads to nonlinearities in the distribution of the permanent magnet's own magnetic chain with the stator current acting through induction to produce ripple torque.

Electromagnetic power influenced by the harmonics of the permanent magnets generated by the UHSPMSM during operation is converted into mechanical power output. Therefore, the ripple torque obtained is:

$$T_{em1} = \frac{P_{e1}}{\omega_m} = -\frac{3}{2}p \sum_{k=1}^{\infty} [(6k+1)\psi_{(6k+1)} + (6k-1)\psi_{(6k-1)}] i_d \sin 6kp\theta \qquad (8)$$

where $P_{e1}$ is the electromagnetic power affected by the harmonics of the permanent magnet, $\omega_m$ is the mechanical angular velocity of the rotor of the UHSPMSM, $T_{em1}$ is the ripple torque, and $\theta$ is the position angle of the rotor surface.

UHSPMSM has a ripple torque of the order 6 k. Its frequency is:

$$f_{t1} = 6k \cdot \left(\frac{pn_r}{60}\right) = 6kf \ (k = 1, 2, 3 \cdots) \qquad (9)$$

where, $f_{t1}$ is the ripple torque fluctuation frequency, $n_r$ is the rotor mechanical speed, rpm, and $f$ is the fundamental wave electric frequency of the UHSPMSM.

(2)　Cogging torque

When the rotor moves relative to the stator tooth and slot structure, there is a large difference in the permeability of the tooth and slot material. This makes the air gap magnetic field no longer an ideal sinusoidal distribution. The cogging torque is generated by the interaction of the air gap magnetic field harmonics affected by stator cogging with the stator input current.

The electromagnetic power generated by the cogging slotting of the UHSPMSM during operation is converted into a mechanical power output. Therefore, the cogging torque obtained is:

$$T_{em2} = \frac{P_{e2}}{\omega_m} = \frac{3}{2}p \sum_{k=1}^{\infty} \psi_{qk} i_q \cos 6kp\theta \qquad (10)$$

where $P_{e2}$ is the electromagnetic power affected by the cogging, and $T_{em2}$ is the cogging torque.

The UHSPMSM has a cogging torque of the order 6 k. Its frequency is:

$$f_{t2} = 6k \cdot \left(\frac{pn_r}{60}\right) = 6kf \quad (k = 1, 2, 3 \cdots) \qquad (11)$$

where $f_{t2}$ is the electromagnetic torque fluctuation frequency.

(3)　Pulsating torque

The nonlinear characteristics of the UHSEAC controller leads to current harmonic components in the UHSPMSM stator. The current fed into the stator has mainly harmonics of the 5th, 7th, 11th, and 13th order. However, due to the property that the higher the order of the current harmonics, the smaller the amplitude, only the 5th and 7th harmonics are generally considered in the analysis process. The pulsating torque is due to the interaction of the stator input current harmonics with the fundamental magnetic field of the permanent magnets.

The three-phase current input to the stator of the UHSPMSM is:

$$i_{abc,h} = \begin{bmatrix} i_{a,h} \\ i_{b,h} \\ i_{c,h} \end{bmatrix} = \begin{bmatrix} i_h \sin(hp\theta + \varphi_h) \\ i_h \sin\left(hp\theta - \frac{2\pi}{3} + \varphi_h\right) \\ i_h \sin\left(hp\theta + \frac{2\pi}{3} + \varphi_h\right) \end{bmatrix} \qquad (12)$$

Electromagnetic power influenced by current harmonics generated by the UHSPMSM during operation is converted into mechanical power output. Therefore, the pulsating torque obtained is:

$$T_{em3} = \frac{P_{e3}}{\omega_m} = \frac{3}{2}p\{\psi_1 i_h \cos[(h-1)p\theta + \varphi_h]\} \tag{13}$$

where $P_{e3}$ is the electromagnetic power affected by current harmonics, and $T_{em3}$ is the pulsating torque.

The pulsating torque is generated by the hth time harmonic in conjunction with the magnetic field of the permanent magnet. Its fluctuation frequency is (h − 1)*f*. The pulsating torque frequency is characterized as:

$$f_{t3} = (h-1)f \quad (h = 1,5,7\cdots) \tag{14}$$

(4)    Electromagnetic torque components

By analyzing the influencing factors generated by electromagnetic torque components, electromagnetic torque components can be expressed as:

$$T_e^* = -\frac{3}{2}p \sum_{k=1}^{\infty} [(6k+1)\psi_{(6k+1)} + (6k-1)\psi_{(6k-1)}]i_d \sin 6kp\theta$$
$$+\frac{3}{2}p \sum_{k=1}^{\infty} \psi_{qk}i_q \cos 6kp\theta + \frac{3}{2}p\{\psi_1 i_h \cos[(h-1)p\theta + \varphi_h]\} \tag{15}$$

where $T_e{}^*$ is the electromagnetic torque components.

The electromagnetic torque components frequency is characterized as:

$$f^* = 6kf + (h-1)f \quad (k = 1,2,3\cdots; h = 1,5,7\cdots) \tag{16}$$

**4. UHSPMSM Electromagnetic Torque Components' Amplitude–Frequency Characteristics**

*4.1. Electromagnetic Torque Components' Characteristics at Different Speeds*

The UHSPMSM used in this study belongs to the type of frequency conversion speed regulation. It is realized by adjusting the base frequency of the stator input current to regulate the speed. The rated speed of the UHSPMSM studied in this paper is 80,000 rpm. The electromagnetic torque component characteristics of the UHSPMSM at 70,000 rpm, 80,000 rpm, and 90,000 rpm will be studied. From the UHSEAC external characteristics' experimental bench, the input current value of UHSEAC at the above rotation speeds can be obtained. In the Maxwell 2D model of the UHSEAC, enter the same current input as in the experiment. The variation of the internal magnetic density cloud diagram of the UHSPMSM at different speeds is shown in Figure 4. When the speed of the UHSPMSM is at 70,000 rpm, 80,000 rpm, and 90,000 rpm, the angles turned through a sampling point are 33.6°, 34.1°, and 34.6°, respectively.

The higher the speed in the same sampling point time, the larger the angle it turns. It can be shown that the base frequency of the stator input current of the UHSPMSM increases, and the rotating magnetic field speed increases. Ultimately, its output speed increases. When the speed modulation of the UHSPMSM is 70,000 rpm, 80,000 rpm, and 90,000 rpm, the fundamental frequency of the current input to the stator is 1166.6 Hz, 1333.3 Hz, and 1500 Hz.

The ideal UHSPMSM is driven by sinusoidal current and generates a standard sinusoidal back-electromotive force [37]. Therefore, there is no torque ripple during operation. However, the stator armature winding current is limited by the inverter capacity and winding inductance, and the back-electromotive force is not a standard sine wave but fluctuates around an ideal waveform. The variation of the back-electromotive force of the UHSPMSM at different speeds is shown in Figure 5. When the speed of the UHSPMSM is 70,000 rpm, 80,000 rpm, and 90,000 rpm, the amplitude of the back-electromotive force is 193.6 V, 221.2 V, and 248.9 V.

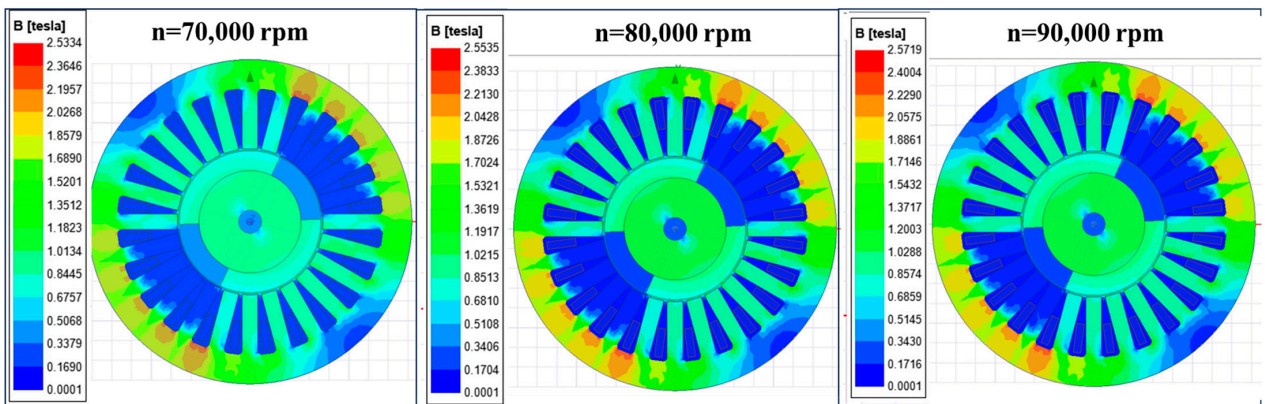

**Figure 4.** Internal magnetic density cloud diagram of the UHSPMSM at different speeds.

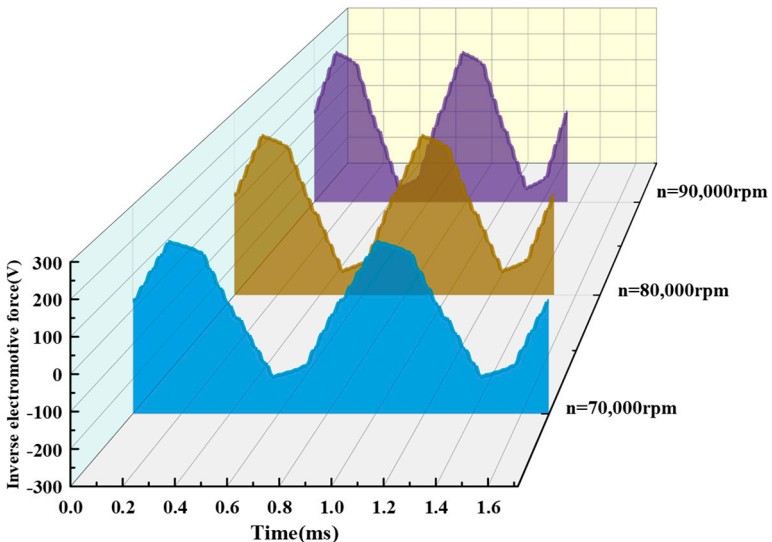

**Figure 5.** Back-electromotive force of the UHSPMSM at different speeds.

Once the structure of the UHSPMSM has been designed, the factor that determines the change in the back-electromotive force is the rotor speed. As the rotor speed increases, the back-electromotive force becomes larger. An increase in the back-electromotive force prevents the input current to the stator armature winding coil, resulting in a lower stator input current. This causes the output electromagnetic torque value to drop and not stabilize at the rated torque. Therefore, in order to keep the UHSPMSM always maintained at the same torque output, it is necessary to increase the amplitude of the stator input current.

The speed control period of the UHSEAC controller is 10 ms. The time-domain diagram of the electromagnetic torque component changes of the UHSPMSM at different speeds is shown in Figure 6. When the speed increases from 70,000 rpm to 80,000 rpm, the amplitude of the 6th order frequency of the ripple torque rises by 0.04 mN·m, the amplitude of the 6th order frequency of the cogging torque is elevated by 7.3 mN·m, and the amplitude of the pulsating torque increases by 6.7 mN·m for the 4th order frequency and 4.2 mN·m for the 6th order frequency. When the speed is increased from 80,000 rpm to 90,000 rpm, the amplitude of the 6th order frequency of the ripple torque rises by 0.05 mN·m, the amplitude of the 6th order frequency of the cogging torque increases by 5.7 mN·m, the amplitude of the 4th order frequency of pulsating torque increases by 10.7 mN·m, and the amplitude of the 6th order frequency increases by 5.4 mN·m.

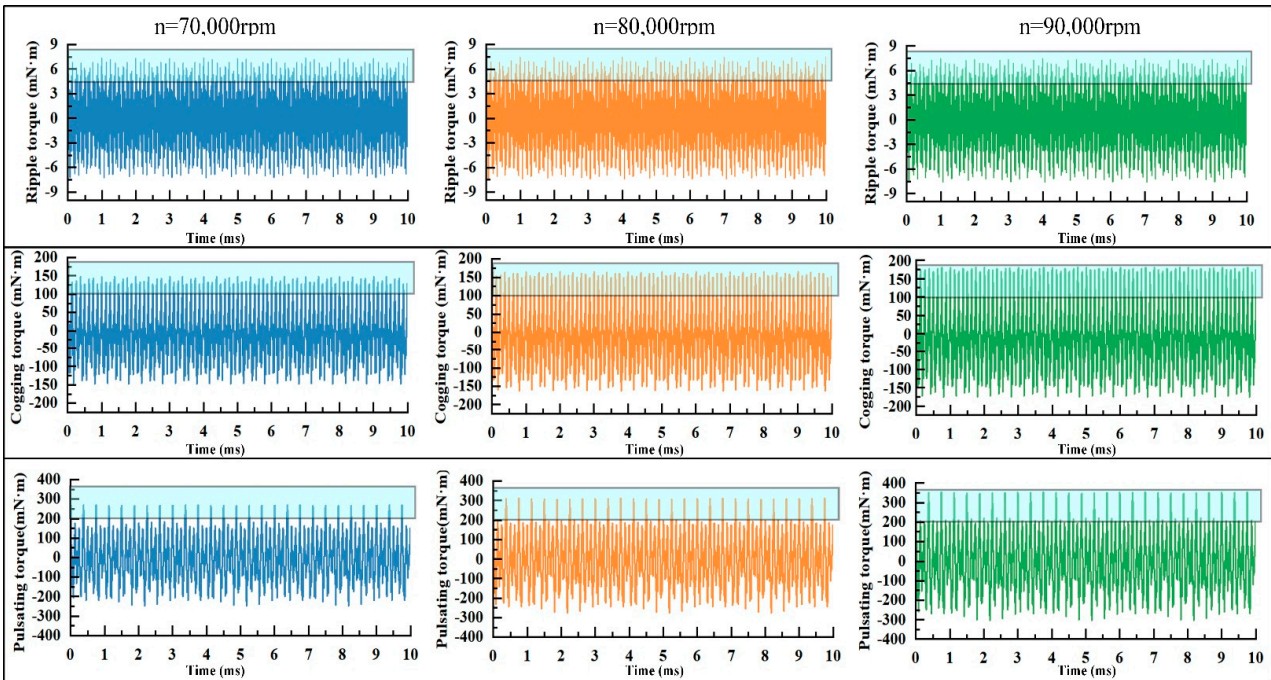

**Figure 6.** Time-domain diagram of electromagnetic torque component changes of the UHSPMSM at different speeds.

When the speed increases from 70,000 rpm to 80,000 rpm, the amplitude of the electromagnetic torque components is increased by 6.7 mN·m for the 4th order frequency and 11.6 mN·m for the 6th order frequency. When the speed is increased from 80,000 rpm to 90,000 rpm, the amplitude of the electromagnetic torque components is elevated by 10.7 mN·m for the 4th order frequency and 11.3 mN·m for the 6th order frequency. The amplitude of the electromagnetic torque components of the UHSPMSM under rated torque becomes larger with the increase of speed. The ripple torque is analyzed to obtain a small amplitude, which accounts for less than 0.2% of electromagnetic torque. The maximum magnitude of the cogging torque at low-order frequency is about 6.5% of electromagnetic torque. The maximum magnitude of the pulsating torque at low-order frequency is about 5.9% of electromagnetic torque. The cogging torque and pulsating torque have a large influence on the torque output stability of the UHSPMSM.

The electromagnetic torque components of the UHSPMSM increase with speed, leading to the torque output being poorly stabilized. This adversely affects the dynamic response performance of the UHSPMSM. In order to verify the frequency and amplitude changes of electromagnetic torque components more accurately, the time-domain plots of ripple torque, cogging torque, and pulsating torque are fast Fourier transformed, as shown in Table 3. When the speed of the UHSPMSM is 70,000 rpm, the base frequency of the stator input current is 1166.66 Hz. The 4th order frequency of the electromagnetic torque components is 4666.66 Hz, and the 6th order frequency is 7000 Hz. When the speed of the UHSPMSM is 80,000 rpm, the base frequency of the stator input current is 1333.33 Hz. The 4th order frequency of electromagnetic torque components is 5333.33 Hz, and the 6th order frequency is 8000 Hz. When the speed of the UHSPMSM is 90,000 rpm, the base frequency of the stator input current is 1500 Hz. The 4th order frequency of the electromagnetic torque components is 6000 Hz, and the 6th order frequency is 9000 Hz.

When the UHSPMS is under different speeds, the frequency of the ripple torque and the cogging torque are the 6th-order current fundamental frequency, and the frequency of the pulsating torque contains the 4th- and 6th-order current fundamental frequency. Thus, the frequency of the electromagnetic torque components contains 4th- and 6th-order current fundamental frequencies. As the speed of the UHSPMSM increases, the

fundamental frequency of the stator input current increases. This leads to an increase in the frequency of the electromagnetic torque components. The electromagnetic torque components are small in amplitude at higher order frequencies and account for less than 1% of the actual output torque. Therefore, electromagnetic torque components at 4th- and 6th-order current fundamental frequencies are worth analyzing.

**Table 3.** Table of electromagnetic torque components' amplitude and frequency values at different speeds.

| Speed (rpm) | Frequency (Hz) | Ripple Torque (mN·m) | Cogging Torque (mN·m) | Pulsating Torque (mN·m) | Electromagnetic Torque Components (mN·m) |
|---|---|---|---|---|---|
| 70,000 | 4666.66 | 0 | 0 | 76.9 | 76.9 |
| | 7000 | 2.1 | 84.9 | 54.8 | 141.8 |
| 80,000 | 5333.33 | 0 | 0 | 83.6 | 83.6 |
| | 8000 | 2.1 | 92.3 | 59 | 153.4 |
| 90,000 | 5333.33 | 0 | 0 | 88.7 | 88.7 |
| | 9000 | 2.2 | 98.1 | 64.4 | 164.6 |

### 4.2. Electromagnetic Torque Components' Characteristics under Different Torques

In the Maxwell 2D model of the UHSPMSM, the basic frequency of the stator input current of the UHSPMSM is maintained at 1333.33Hz to maintain the speed at the rated speed of 80,000 rpm. By changing the effective value of the stator input current, the output electromagnetic torque of the UHSPMSM is modulated to 1.3 N·m, 1.5 N·m, and 1.7 N·m. An air gap flux density diagram of the UHSPMSM under different torques is shown in Figure 7. When the output electromagnetic torque of the UHSPMSM is 1.3 N·m, the radial air gap flux density amplitude is 1.1 T, and the tangential air gap flux density amplitude is 0.4 T. When the output electromagnetic torque is 1.5 N·m, the radial air gap flux density amplitude is 1.2 T, and the tangential air gap flux density amplitude is 0.5 T. When the output electromagnetic torque is 1.7 N·m, the radial air gap flux density amplitude is 1.3 T, and the tangential air gap flux density amplitude is 0.5 T.

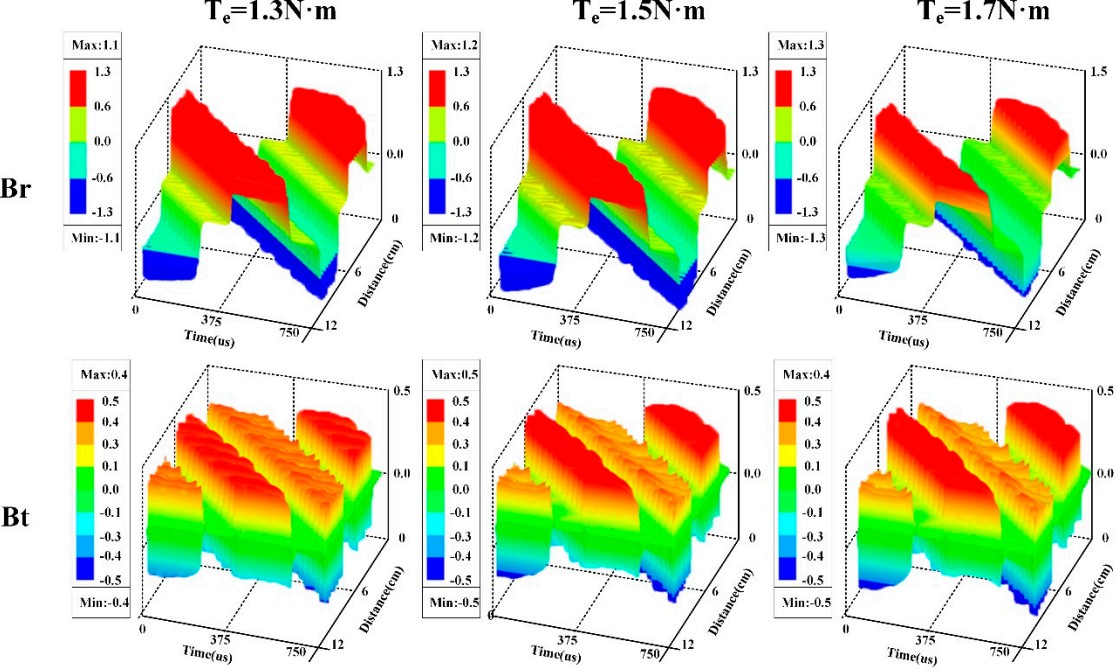

**Figure 7.** Air-gap flux density diagram of the UHSPMSM under different torques.

The air gap flux densities at the three different electromagnetic torques are all non-sinusoidal variations. They include the harmonic component generated by permanent magnet field harmonics, stator slotting, and current harmonics. When the stator input current increases, the rotating magnetic field in the UHSPMSM is enhanced. When the stator input current increases, the rotating magnetic field generated by the stator circular rotating magnetic potential and the main magnetic field of the rotor permanent magnets are enhanced. The air gap magnetic field synthesized during synchronous operation is enhanced. Therefore, the electromagnetic torque produced is increased.

A time-domain plot of the electromagnetic torque components of the UHSPMSM with an electromagnetic torque component output of 2 electrical cycles was created. In order to study the electromagnetic torque component variations more accurately, the time-domain map of the electromagnetic torque components is obtained as a frequency-domain map by fast Fourier variation. An electromagnetic torque components' diagram of the UHSPMSM under different torques is shown in Figure 8. The speed of the UHSPMSM is fixed at 80,000 rpm, and the stator input current fundamental frequency is 1333.33 Hz. When the electromagnetic torque is increased from 1.3 N·m to 1.5 N·m, the 6th-order frequency amplitudes of ripple torque and cogging torque are elevated by 0.1 mN·m and 13.8 mN·m, the amplitude of the 4th-order frequency of the ripple torque is elevated by 11.5 mN·m, and the amplitude of the 6th-order frequency is elevated by 6.5 mN·m. The amplitude of the 4th-order frequency of the electromagnetic torque components increases by 11.5 mN·m, and the amplitude of the 6th-order frequency increases by 19.3 mN·m when the electromagnetic torque is increased from 1.3 N·m to 1.5 N·m. When the electromagnetic torque is elevated from 1.5 N·m to 1.7 N·m, the amplitude of 6th-order frequency of the ripple torque and cogging torque is elevated by 0.1 mN·m and 12.0 mN·m, the amplitude of the 4th-order frequency of the ripple torque increases by 10.7 mN·m, and the amplitude of the 6th-order frequency increases by 9.4 mN·m. When the electromagnetic torque is increased from 1.5 N·m to 1.7 N·m, the amplitude of the 4th- and 6th-order frequencies of the electromagnetic torque components are increased by 10.7 mN·m and 21.7 mN·m, respectively.

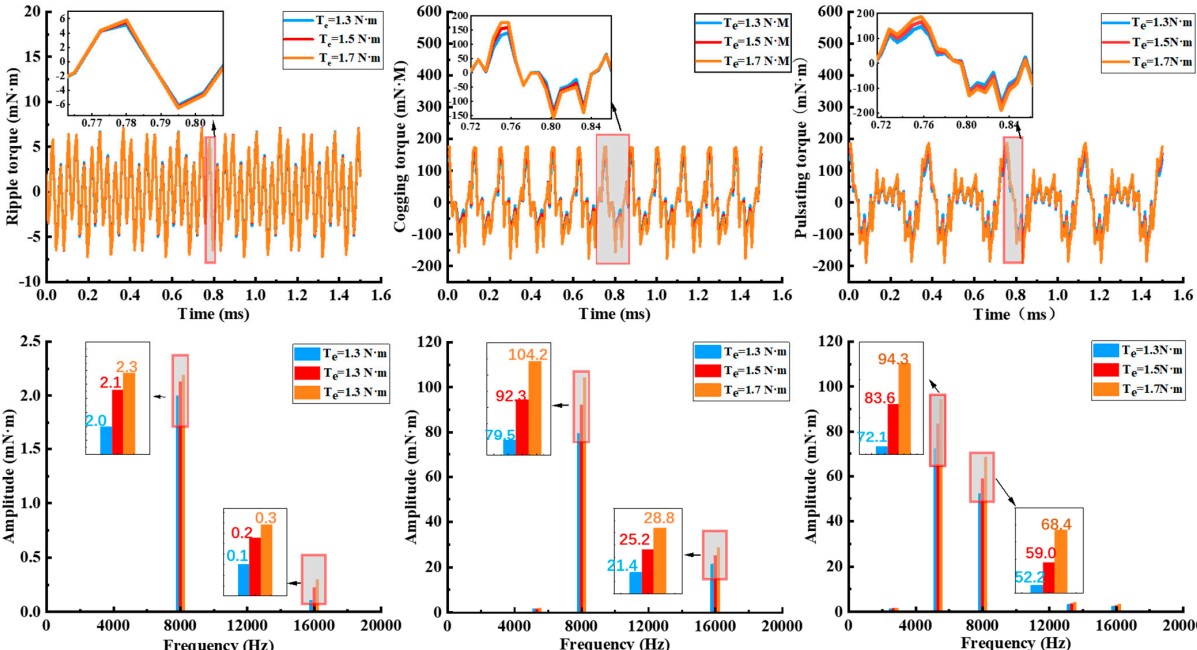

**Figure 8.** Diagram of the electromagnetic torque components of the UHSPMSM under different torques.

Ripple torque and cogging torque are at 6th-order current fundamental frequency, and pulsating torque is at 4th- and 6th-order current fundamental frequency. Therefore, the frequency of the electromagnetic torque components contain 4th- and 6th-order current fundamental frequencies. When the speed is 80,000 rpm, the 4th-order frequency is 5333.33 Hz and the 6th-order frequency is 8000 Hz. Electromagnetic torque components are smaller in amplitude at higher-order frequencies and account for a smaller percentage of the electromagnetic torque. It is sufficient to analyze only the ripple torque, cogging torque, and pulsating torque, which have larger amplitudes at lower-order frequencies. It is obtained from the analysis that the ripple torque accounts for less than 0.2% of the electromagnetic torque, the amplitude of the cogging torque is about 6.0% of the electromagnetic torque at the lower-order frequency, the amplitude of the pulsating torque at the 4th-order frequency accounts for about 5.5% of the electromagnetic torque, and the amplitude at the 6th-order frequency accounts for about 4.0% of the electromagnetic torque.

## 5. Conclusions

There are a simulation comparison and a theoretical description of different sources of pulsation factors. This paper extracts the electromagnetic excitation of UHSPMSM and analyzes the characteristic changes of the electromagnetic excitation of the UHSPMSM under different operating states. Several conclusions can be summarized as follows:

(1) At rated electromagnetic torque, the speed of the UHSPMSM increases, requiring an increase in the input current fundamental frequency. However, an increase in the current fundamental frequency increases the rate of change of the air gap magnetic field, leading to an increase in the electromagnetic excitation frequency. As the speed increases, the back-electromotive force increases. It is necessary to increase the input current amplitude to maintain it at the rated electromagnetic torque. This results in an increase in the amplitude of the electromagnetic excitation. When the speed increases from 70,000 rpm to 90,000 rpm, the 4th-order frequency of the electromagnetic excitation increases by 1333.33 Hz and the amplitude increases by 11.8 mN·m, and the 6th-order frequency increases by 2000 Hz and the amplitude increases by 22.8 mN·m.

(2) At the rated speed, the electromagnetic torque of the UHSPMSM rises, requiring an increase in the input current amplitude. The increase in current amplitude increases the air gap magnetic field strength. This leads to a consequent increase in the electromagnetic excitation amplitude. The speed is constant, the input current fundamental frequency is constant, and the electromagnetic excitation frequency is constant. When the electromagnetic torque increases from 1.3 N·m to 1.7 N·m, the electromagnetic excitation 4th-order frequency remains unchanged at 5333.33 Hz and the amplitude increases by 22.2 mN·m, and the 6th-order frequency remains unchanged at 8000 Hz and the amplitude increases by 41.1 mN·m.

This paper constructs the observation object of the high-frequency state observer and conducts the preliminaries for the design of the UHSEAC controller. The amplitude–frequency characteristic law of electromagnetic excitation can provide reference opinions for the structural design optimization of UHSPMSM. It can also provide theoretical support for the control method and analysis of the stability of UHSEAC.

**Author Contributions:** Conceptualization, J.Z. (Jiaming Zhou) and D.H.; methodology, J.Z. (Jiaming Zhou) and J.S.; software, J.Z. (Jinming Zhang) and G.W.; validation, J.Z. (Jiaming Zhou), J.Z. (Jinming Zhang) and F.Y.; formal analysis, C.Z.; investigation, Y.L.; resources, D.H. and G.W.; data curation, Z.Z.; writing—original draft preparation, J.Z. (Jiaming Zhou); writing—review and editing, D.H.; visualization, J.S. and G.W.; supervision, D.H.; project administration, D.H.; funding acquisition, D.H. All authors have read and agreed to the published version of the manuscript.

**Funding:** This research is supported by Weifang University of Science and Technology High-level Talent Research Start-up Fund Project (KJRC2023001), the National Key Research and Development

Program of China (2022YFE0137600), Weifang University of Science and Technology 2023 School-level Project (2023KJ02 and 2023KJ03).

**Data Availability Statement:** The data presented in this study are available on request from the corresponding author. The data are not publicly available due to third-party confidentiality agreement restrictions.

**Conflicts of Interest:** The authors declare no conflicts of interest.

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
