# Peer review of "Electromagnetic Torque Components Analysis of Ultra-High-Speed Permanent-Magnet Synchronous Motor for Fuel Cell Air Compressor"

_actuators, doi:10.3390/act13050184_

Round 1
Reviewer 1 Report
Comments and Suggestions for Authors
The current paper analyses the operation of an ultra-high-speed permanent magnet synchronous motor driving an ultra-high-speed electric air compressor, with focus on torque behaviour, which is experimentally acquired for different speeds. Furthermore, the significant components of the motor torque such as: cogging, ripple and pulsating are deduced. A comprehensive reference list is provided.
Nevertheless, corrections are required, as follows:
- Remove the first paragraph form the introduction section (it remained from the template)
- The word “scholar” is overused in chapter 1 – please find a suitable synonymous
- Keep an unitary font throughout the paper: check page 7 from line 231 to 242
- Clarify why, instead of the number of pole pairs (p) the term polar logarithm is used
- Figure 3 shows the electromagnetic torque variation for three speed values, as a comparison between experiments and simulation, but no additional information is provided about the simulation software/environment
- In relation to the above comment, Figure 4 shows the Magnetic density cloud inside the UHSPMSM – again no details are given about the software/environment used to plot the figures; the same comment is available for figure 7
- Check reference 25 – page 14 line 477 for typing errors
Comments on the Quality of English LanguagePlease check the spelling for grammar disagreements such in the phrase from page 13 line 403
Reviewer 2 Report
Comments and Suggestions for Authors
The aim of this paper is to analyze by simulations an experiments the electromagnetic torque components of an ultra-high-speed surface-mounted permanent-magnet (UHS-SMPM) synchronous motor drive for use in fuel-cell air-compressor applications. Although the paper provides a significant research effort from the authors, it reveals several shortcomings of technical editing seemingly due to insufficient familiarity of the authors with PM synchronous-motor electromagnetic analysis, so that the paper has to be thoroughly rewritten, at least, in following terms: (i) the paper title is not well-suited to the paper main contribution, an alternative title being suggested as ’Electromagnetic torque analysis of an ultra-high-speed permanent-magnet synchronous motor for fuel-cell air-compressor drive’ (which is also in logical sequel with the title of previous published paper (in 2023, not 2022) of the authors, given as paper reference [6]); (ii) the first paragraph of the introduction is useless, and thus can be removed; (iii) the content of Figure 1 has to be reconceived in terms of ‘electromagnetic torque components’ instead of ‘electromagnetic excitation extraction’, which is meaningless; (iv) data of Tables 1 and 2 have to be completed with more details concerning the rotor magnets used, i.e. they are only two (symbol, p =1, means ’one pole-pair’, not ‘polar logarithm’ (?!), surface-mounted, diametrically (or radially ?) magnetized, and made of NdFeB; (v) the harmonics order of the electromagnetic torque ripple and of the cogging torque must be theoretically justified; (vi) how the quasi-trapezoidal back EMF of the motor matches the sine-wave-controlled electrical current stator supply ?; (vii) the text and figure captions of paper Section 4 are confused, even, incorrect, and have to be reconsidered; (viii) Figure 5 is trivial, since the back EMF, by definition, is proportional to the rotor speed; (ix) the English writing of the paper has to be substantially improved.
Comments on the Quality of English LanguageEnglish writing of the paper has to be substantially improved.
Reviewer 3 Report
Comments and Suggestions for Authors
The manuscript entitled: “Electromagnetic excitation extraction and analysis of ultra-high-speed permanent magnet synchronous motor" concerns the analysis of the performance of the motor in selected operation states. The manuscript can be interesting to research interested in minimization of the torque ripples in PMSM.
The theoretical part of the manuscript is substantial and prepared with a model description. I will have minor remarks to develop a mathematical model. The manuscript cites 33 reference positions, but in my opinion, the introduction should be enriched with three-four manuscripts concerned about different methods of minimization torque ripples in different types of synchronous motors. In my opinion, the result sections should be extended to an interesting comparison assessment of the influence of various harmonics for different loading torques, with interesting charts.
The detailed comments to the authors are the following:
1. In the abstract, the authors inform the readers of the "electromagnetic torque model of UHSPMSM", but with a more detailed description of the developed model.
2. Rows 29 - 36 include the part of the Actuators template for authors and must be removed.
3. In row 46 the authors wrote: "(PMSM) consists of three parts: mechanical system, electronic control system and coupled magnetic field", but it is not true. This is because the motor includes a wounded stator and rotor equipped with permanent magnets. The drive can include a motor, gear and electronic motion controller. This sentence must be improved before the final publication.
4. In row 62 the authors wrote "Peng et al [23] investigated the effect of rotor permanent magnet width on torque pulsation". The electromagnetic torque generated by synchronous machines contain the synchronous torque modified by cogging torque - from theory of electrical machines. There are many manuscript focus on the minimization and many approach: (a) "Research on Synergistic Reduction of Cogging Torque and Ripple Torque of Interior Permanent Magnet Synchronous Motor Based on Magnetic Field Harmonic Offset Method", (b) "Optimization of the synchronous motor with hybrid permanent magnet excitation system", and (c) "A Cogging Torque Minimization Procedure for Interior Permanent Magnet Synchronous Motors Based on a Progressive Modification of the Rotor Lamination Geometry".
5. The "Polar arc coefficient" in Table 1 should be replaced by the magnet span - for discussion.
6. In rows 109 and 110 the symbols of magnetic axis d and q should be written in italic font.
7. In row 116 the commas between phase currents ia, ib and ic look as in different styles.
8. In row 120 authors use the name " winding magnetic chains" for total magnetic fluxes coupled in the phases winding. In electrical machines, these winding magnetic chains are named fluxes in voltage equation. The authors should consider resignation from these names.
9. It is a lack of description of mutual (M) and self-inductances in equation (2).
10. In rows 131 and 132 the d and q should be written by italic font.
11. In Table 2 the "Permanent Magnet Chain", the words magnet and chain should be written in normal letters.
12. The quality of Figure 3 is poor and must be improved before final publication.
13. Under equations (8) and (9) authors underline the unit for Pel [w], T [Nm], and [Hz] in my opinion the authors should resign from the unit. This is because the readers of Actuators know the unit of frequency, torque and electrical power.
14. The quality of Figure 4 is poor and must be improved. It is difficult to see the magnetic flux density scale.
15. The reverse electromotive force should be replaced by back-electromotive force. It is more correct from electrical machines theory.
16. The quality of Figure 6 is poor and must be improved before final publication.
17. The quality of Figure 7 and Figure 8 is poor and must be improved before final publication.
18. The authors should underline the novelty and new theory/approach to the manuscript. There is a simulation comparison and theoretical description of different sources of pulsation factors.
19. The article should be equipped with interesting graphic (charts) comparisons for various scenarios of research.

The English grammar modification are minor.
Round 2
Reviewer 2 Report
Comments and Suggestions for Authors
This revised paper – in which the authors analyze by simulations and experiments the electromagnetic torque components of an ultra-high-speed surface-mounted permanent-magnet (UHS-SMPM) synchronous motor drive for use in fuel-cell air-compressor applications – provides the authors’ pertinent effort in responding to reviewer’s requests by (i) changing the paper title, as suggested, to make it well-suited to the paper content; (ii) reconceiving in terms of ‘electromagnetic torque components’ the content of Figure 1, as well as the text and figure captions of paper Section 4; (iii) completing the data of Tables 1 and 2; (iv) justifying the results displayed on Figure 5, as well as the harmonics of the electromagnetic torque ripple and of the cogging torque; (v) substantially improving the English writing of the paper. Hence, there are no other requests to be addressed to the authors.
Reviewer 3 Report
Comments and Suggestions for Authors
The author provide good quality discussion and made good modifications in the manuscript body. The major part of comments are taken into account during improvements of the manuscript.
The one point to discussion can be:
1. In the Table 1 the authors give readers information "Surface mounted" are applied, whereas on the Figure 1 are presented the buried permanent magnet rotor.
2. The reference list should be minor modified to adapted to MDPI requirements, e.g. "Ma C.; Shi H.; Nie P,.; Wu J. "
Manuscript in current form manuscript can be accepted to publication.
The minor correction can be made during the correction by English Editor.
